# Effects of Different Nutritional Patterns and Physical Activity on Body Composition: A Gender and Age Group Comparative Study

**DOI:** 10.3390/foods13040529

**Published:** 2024-02-08

**Authors:** Mauro Lombardo, Alessandra Feraco, Elisabetta Camajani, Stefania Gorini, Rocky Strollo, Andrea Armani, Elvira Padua, Massimiliano Caprio

**Affiliations:** 1Department of Human Sciences and Promotion of the Quality of Life, San Raffaele Open University, Via di Val Cannuta, 247, 00166 Rome, Italy; alessandra.feraco@uniroma5.it (A.F.); elisabetta.camajani@uniroma5.it (E.C.); stefania.gorini@uniroma5.it (S.G.); rocky.strollo@uniroma5.it (R.S.); andrea.armani@uniroma5.it (A.A.); elvira.padua@uniroma5.it (E.P.); massimiliano.caprio@uniroma5.it (M.C.); 2Laboratory of Cardiovascular Endocrinology, San Raffaele Research Institute, IRCCS San Raffaele Roma, Via di Val Cannuta, 247, 00166 Rome, Italy

**Keywords:** food preferences, cross-sectional studies, dietary patterns, taste, diet, vegetarian, body composition

## Abstract

This cross-sectional study analyses differences in dietary habits, taste preferences, variety of protein sources and body composition (BC) profiles among individuals following omnivorous, flexitarian, lacto-ovo-vegetarian and pescatarian diets. Furthermore, it assesses the correlations between these dietary patterns and various sports, classified by exercise intensity, in relation to BC parameters. The study analysed the eating habits and BC data of 1342 participants aged 18–65 years, classified into four diet groups based on their 7-day food diaries and questionnaire responses. Our analysis revealed gender- and age-related differences in weekly food consumption and protein source variety, with men generally consuming more meat, processed meat and fish than women, especially in younger age groups. Differences in dairy and soy consumption were also noted between age groups, while legume and soy preferences showed no gender disparity across all ages. Among non-sporting individuals, vegetarians exhibited lower fat mass (FM%) compared to other diets, while among athletes, vegetarians and pescatarians in in endurance and strength sports, respectively, displayed lower FM%, with flexitarians and omnivores in endurance sports showing higher FM%. Non-athletic omnivores and vegetarians demonstrated a greater proportion of body protein, while among athletes, those engaged in strength training exhibited a higher body protein content across all dietary groups compared to those in endurance training. Among non-athletic groups, vegetarians exhibited the lowest FM/FFM (fat mass/fat-free mass) ratio, while among athletes, vegetarians in endurance sports and participants in strength training across other diets showed lower FM/FFM ratios. The results emphasise the complex interaction between diet, BC and lifestyle choices, revealing how different combinations of diet and sport are associated with optimised BC.

## 1. Introduction

A varied diet is crucial for maintaining optimal long-term health and reducing the risk of developing chronic diseases [1]. Consuming a diversity of foods not only provides a broad spectrum of essential nutrients, but also reduces the risk of obesity and metabolic alterations related to body fat expansion [2]. Nutritional diversification can have beneficial effects on the immune system, metabolic function and general health [3]. In particular, a varied diet has been associated with a reduced risk of heart disease, especially if plant-based proteins are included [4,5,6]. Plant-based protein dietary patterns, which emphasise healthy food consumption in particular, are helpful in reducing the risks of major chronic diseases, including cardiovascular disease [7], type 2 diabetes [8], cancer [9] and premature death.

Dietary models, such as flexitarian, omnivore, pescatarian and vegetarian, provide diverse approaches to nutrition that reflect varying ethical, environmental and health motivations [10]. Each diet has advantages and potential risks, for example, from the occasional inclusion of fish and meat for flexitarians, to the complete exclusion of meat for vegetarians [11]. There are few studies on the effects of these different models on body composition (BC) and eating habits. These choices could significantly influence diet quality, eating disorder risks and health [12]. In addition, the interaction between dietary patterns and physical activity plays a crucial role in BC. Understanding how these dietary patterns interact with physical activity may be crucial for health and dietary recommendations [13,14].

The objective of this cross-sectional research is to investigate differences in dietary habits, taste preferences and variety of protein sources, as well as body composition (BC) profiles among individuals following various diets, including omnivorous, flexitarian, lacto-ovo-vegetarian and pescatarian. The study also sought to assess correlations between different dietary patterns and various sports, classified by exercise intensity, in relation to BC parameters.

## 2. Methods

### 2.1. Subjects

In this cross-sectional study, we examined a cohort of participants, including adults and adolescents, who attended a medical centre specialising in nutrition and metabolism in Rome, Italy. Patient enrolment began in May 2023 and, as of November 2023, we recorded a total of 1500 surveys and medical visits. We included a wide age range to obtain a comprehensive view of dietary patterns across age groups. The clinical centre primarily caters to individuals seeking expert assessment and guidance for their BC and related health concerns. Consequently, the study population exhibited unique characteristics that may deviate from the broader demographics of the country. Notably, the individuals participating in this study showed a heightened interest in and awareness of BC, which could potentially have influenced their dietary choices and physical activity habits. Moreover, the medical centre’s specialized emphasis on BC assessment could have attracted individuals with specific health needs or fitness aspirations. While participants in this study are patients from a medical centre, seeking lifestyle modifications, it is essential to clarify that their initial health objectives do not influence the evaluation of their dietary habits and BC in the context of our study. This approach allows for an unbiased assessment of the impact of varied protein sources and physical activity on BC.

The screening process involved excluding subjects who were pregnant or breastfeeding, those using medications known to affect weight (such as glucocorticoids, oestrogen, and anticonvulsants) and individuals with specific medical conditions like alcoholism or chronic kidney disease. At the conclusion of the data analysis process, we excluded 26 individuals from the final analysis. This exclusion was necessary due to missing or incomplete data. We have also excluded the data of patients aged under 18 or over 65 in order to make the sample more homogeneous.

The study’s procedures, including the consent form, were reviewed and approved by the IRCCS San Raffaele ethics committee (registration number RP 23/13), ensuring compliance with the ethical standards outlined in the Declaration of Helsinki and its subsequent amendments.

### 2.2. Dietary Patterns

In our study, we classified the cohort of participants into distinct dietary groups based on their 7-day food diary and questionnaire responses. This combination of structured questionnaire data and detailed diary entries allowed for a robust analysis of dietary patterns. It provided both quantitative and qualitative insights into the participants’ daily food consumption, contributing significantly to our understanding of the impact of dietary choices on BC and physical activity levels. The categories were as follows: (1) Lacto-ovo-vegetarians: participants in this group reported consuming no meat or fish. This category was defined by a strict adherence to a diet that excluded all forms of meat and seafood, focusing instead on plant-based foods; (2) Pescatarians: this group included individuals who consumed fish but abstain from meat; (3) Omnivores: this group represents a more traditional diet that includes a wide variety of animal proteins, with no specific restrictions on the intake of meat and fish; (4) Flexitarians: this group was defined by a limited consumption of meat and fish, with an intake of both no more than once a week. It represents a more flexible approach to diet, incorporating elements of both vegetarianism and omnivorous habits, but with a conscious effort to limit meat and fish consumption. This classification allowed us to effectively analyse and compare eating habits and health outcomes between different dietary preferences, providing valuable insights into the impact of different dietary patterns. We also analysed the range of protein sources—meat, processed meat, fish, eggs, dairy, legumes and soy—according to the food diary. Diversity was rated according to the presence of each protein source, from 0 to 7, with higher scores indicating greater diversity.

### 2.3. Questionnaire 

Participants were asked to complete a comprehensive questionnaire prior to their initial visit to the medical centre. The questionnaire was administered electronically, accessible via any device with an Internet connection, and took, on average, approximately 30 min to complete. Participants were given the opportunity to give or withhold their consent to participate in the study when accessing the survey link. The online survey was structured for self-administration and, although a formal validation test was not conducted, its format was consistent with other validated food taste questionnaires [15]. To ensure anonymity, all responses were collected without personal identifiers. The questionnaire, divided into four sections, had already been used in our previous studies [16,17]. In this study we have focused on the questionnaire parts about food preferences and inclination for sport. Subjects’ taste preferences for specific foods were investigated through the question “Do you like the following foods?” with response options “I like”, “I don’t like” and “I don’t know”. The foods included were cow’s milk, plant-based alternatives such as soya milk, low-fat and low-sugar yoghurt, fresh cheese, various types of meat, processed meats such as ham, fish, eggs, legumes, cooked and raw vegetables, fruit, cereals such as spelt and barley, foods containing wholemeal flour, nuts, tofu (soya) and dark chocolate with a cocoa content of more than 75%. The questionnaire also included questions on sporting activity: “Do you play a sport?” “Which sport?”, “When?”, and “How many hours (per week)?”. In our study, the term ‘sporting activity’ refers to participants’ involvement in various sports. We inquired about the types of sports they engaged in to assess the impact of these specific physical activities on their dietary patterns and body composition. This detailed inquiry into sports participation is crucial for understanding the nuanced relationship between exercise types and dietary habits. To simplify the calculations, we categorised all sports into four main groups: Endurance Sports, Skill Sports, Strength Training, and Team Sports (Appendix A).

### 2.4. Body Composition

Body weight was measured with a calibrated electronic scale from the BIA TANITA Corporation, Sportlife, Tokyo, Japan. This scale, with a range of 0–200 kg and an accuracy of 100 g, was used while the participants, who had been fasting for at least one night, were dressed only in their underwear [18]. Height was measured with a stadiometer, ensuring that participants stood with their heels, buttocks, shoulders and head in contact with a vertical bar. Body mass index (BMI) was calculated as weight in kilograms divided by height in square metres. The abdominal circumference (AC) was measured at the midpoint between the iliac crest and the lowest rib, with participants in a standing position and during minimal breathing. With regard to BC, fat mass (FM), fat-free mass (FFM) and hydration status (TBW) were recorded using the BIA Tanita BC-420 MA, a validated instrument compared to BodPod [18,19]. Measurements were scheduled at least three hours after waking up, three hours after meals and twelve hours after any strenuous exercise. Participants were instructed to avoid alcohol and, if appropriate, to schedule the examination outside the menstrual period, at least 12 h in advance. In addition, they were asked to urinate before the measurement. To ensure accuracy, averages of two independent measurements were taken for all parameters, thus minimising potential errors in the BC analysis.

### 2.5. Statistical Analysis

In this study, we employed a comprehensive statistical approach using SPSS v. 28 (IBM Corporation, Armonk, NY, USA) to analyse a broad spectrum of dietary data from a cohort of 1397 individuals. Descriptive statistics were utilised to delineate the patterns of food consumption and taste preferences, with a specific focus on variations between male and female participants. Chi-square statistics were instrumental in highlighting significant gender-based differences, adhering to a threshold of *p* < 0.05 for statistical significance. Our analysis extended to examining dietary consumption patterns encompassing meat, processed meat, fish, eggs, dairy, legumes and soy. Due to the count-based and non-normal nature of this consumption data, we employed non-parametric tests. Specifically, the Mann–Whitney U test was used for gender comparisons, and the Kruskal–Wallis test was used to discern differences across various age groups and BMI categories. Additionally, ANOVA was conducted to probe into the nuances of continuous variables and their statistical differences between groups. Two-proportion z-tests were used to determine the *p*-values for the comparison of affirmative response rates between genders for each food item. Further, we examined the impact of various diet and sport combinations on the FM/FFM ratio across genders. Data were collected on individuals’ dietary habits, sports activities, and body composition measurements. Statistical analysis, including Chi-square and Fisher’s Exact tests, was performed to determine significant differences in FM/FFM ratio distributions among different diet and sport combinations. We considered *p*-values below 0.05 as indicative of statistically significant differences.

## 3. Results

The study population included 553 males (41.2%) and 789 females (58.8%) (Table 1). The mean age was 39.4 ± 11.8 years for the total cohort, with males younger (38.2 ± 11.6 yrs) than females (40.2 ± 11.8 yrs). Most participants were in the ‘25.0–29.9’ (42.3% of the total) and ‘18.5–24.9’ (31.2%) BMI categories. Table 1 shows the significant gender differences in BC parameters.

The analysis reveals different patterns in BMI distribution between diets and between genders (Table 2). With regard to the Flexitarian diet, the majority of the females fell into the categories of normal weight and overweight, while the males were predominantly in the same categories. The overall trend in this diet group showed a similar distribution, with the highest percentages of normal weight and overweight. In the omnivorous diet group, females fell predominantly into the overweight and normal weight categories, while males fell predominantly into the overweight and obese class I categories. The total group followed this trend, with the majority of participants being overweight and normal weight. Participants on a pescatarian diet showed a majority in the normal weight and overweight categories. This pattern was consistent with the total group, where the highest percentages were in the normal weight and overweight categories. With regard to the vegetarian diet, females belonged mainly to the overweight category, followed by the normal-weight category, while males mainly fell into the overweight and obesity class I categories. The combined group for this diet was predominantly in normal-weight and the overweight categories.

Our analysis revealed notable differences in weekly food consumption data and weekly variety of protein sources between males and females (Table 3) across various age groups (18–30, 31–45 and 46–65 years). For this analysis, information was extracted from 7-day food diaries to determine the weekly frequency of consumption for each protein source. The comparison between different groups was based on the percentage of consumption exceeding the 75th percentile. In the 18–30 age group, significant gender differences were observed in the consumption of meat (*p* < 0.001), processed meat (*p* < 0.001) and fish (*p* = 0.0055), with men consuming these more frequently than women. Dairy consumption was found to be higher in the group of young males. Soy consumption was higher in the younger groups than in the 46–65 group. For the 31–45 age group, meat (*p* = 0.006), processed meat (*p* < 0.001), and fish (*p* = 0.0011) showed significant differences, as in the younger age group. Egg consumption was observed to be highest in men within the 18 to 30 and 46 to 65 age groups. Additionally, the number of protein sources (*p* = 0.866) was not significantly different in the younger while in the oldest age group, significant differences were observed in fish consumption (*p* = 0.0009) and the number of protein sources (*p* = 0.0033), with men showing a higher frequency and variety in these categories. The preference for legumes and soy showed no gender differences in all age groups. 

In the quantitative exploration of the diversity of protein sources within the specified dietary patterns (Figure 1), a significant discordance was identified between the different diets (ANOVA *p* < 0.001). The omnivorous diet was associated with the greatest variety of protein sources, in contrast, the vegetarian diet showed the least variety. No significant gender differences were observed within the dietary categories, with *p*-values of 0.074 for pescatarians, 0.938 for omnivores, 0.762 for flexitarians and 0.887 for vegetarians.

The investigation of food consumption patterns in the flexitarian, omnivore, pescatarian and vegetarian diets revealed distinct preferences and avoidances that align with the dietary restrictions inherent in each regime (Figure 2). The results of the Chi-Squared test indicate significant differences in the consumption of cow’s milk, vegetable drinks, fresh cheese, meat, red meat, processed meat, fish, eggs and tofu. In contrast, food such as low-fat white yoghurt, legumes and various vegetables and cereals showed no significant differences in consumption between diets. 

In the analysis of dietary preferences across four distinct diets—Omnivorous, Pescatarian, Vegetarian, and Flexitarian—we observed notable gender-based differences in food choices (Figure 3). Men, particularly within the Omnivorous and Flexitarian groups, exhibited a significantly higher preference for meat, red meat and processed meats than women, as evidenced by the low *p*-values. This trend in preference for meat and its variants among men was consistent, except in diets that are strictly vegetarian or pescatarian. On the other hand, women, regardless of their dietary pattern, showed a stronger inclination towards plant-based options. This was particularly evident in their preference for cooked vegetables and whole grains, with the differences being statistically significant. The preference for eggs also stood out, with women across all dietary patterns favouring them more than men.

Our investigation of the interaction of dietary habits and sporting activities on BC yielded interesting results. The data, represented in Figure 4, reveal distinct patterns in average FM percentage (FM%), body protein content and fat-free mass ratio (FM/FFM) across a spectrum of diet and sport categories. FM% was found to be lower among non-sporting vegetarians compared to other non-sporting diet groups. Among athletes, vegetarians practising endurance sports and pescatarians engaged in strength sports exhibited lower FM%. Higher FM% was observed in flexitarians participating in endurance and team sports, and among omnivores involved in endurance sports. Across various groups, strength sports appeared to be correlated with a lower FM%, except in the case of vegetarians. Among non-athletes, omnivores and vegetarians displayed higher body protein levels, while among athletes, strength training showed greater effectiveness in increasing body protein compared to endurance training across all four dietary groups. Among non-athletes, the FM/FFM ratio was similar across various diet groups, with vegetarians showing the lowest values, while among athletes, vegetarians practising endurance sports had the lowest FM/FFM ratios. In the other three diet groups, athletes engaged in strength training exhibited lower FM/FFM ratios compared to those in endurance sports.

## 4. Discussion

The present study is part of a growing research context focused on the importance of dietary patterns and physical activity in modulating and preventing chronic diseases [20,21,22,23]. Our study showed significant gender differences in food consumption patterns across various age groups (18–30, 31–45, 46–65), revealing the complex interaction between gender, age and eating habits [24]. Men generally consume more meat, processed meat and fish, with this trend being particularly pronounced in the younger age groups. The 31–45 age group continued to show significant gender differences in the consumption of animal-based foods, potentially influenced by cultural norms associating meat with masculinity [25]. However, in the 46–65 age group, these gender differences became less pronounced. This is in line with previous studies [26,27,28] suggesting a cultural and psychological association of meat consumption with masculinity, strength and vitality among men. Complementing these findings, a recent study [29] provided fascinating insights into sensory perception and attitudes towards meat. It was observed that the hedonic evaluation of meat progressively decreases with increasing redness and intensity of meat flavours, a tendency particularly pronounced among female participants. Women, in stark contrast to men, showed significantly lower hedonic scores for redder meat varieties, such as ostrich, lamb and beef. Men, in contrast, showed higher attitudinal support for red meat. Egg consumption also varied, with significant differences observed in the 18–30 and 46–65 age groups, indicating a gender preference that varies with age. Dairy consumption showed gender differences in the younger group (18–30) but not in the older age groups (31–45, 46–65). Interestingly, no significant gender differences were found for legumes and soy in all age groups, suggesting a more uniform acceptance of these protein sources among both sexes. This could be attributed to the increasing health awareness and popularity of plant-based diets [30]. The only noticeable difference in the number of protein sources was found in the oldest age group (46–65 years), suggesting that dietary diversity in protein sources may become more important in old age. Overall, these trends underline the importance of considering gender and age when developing nutritional recommendations and interventions. The findings also invite further exploration of cultural and psychological factors influencing food choices and the implications of these choices for long-term health [27,28]. The dietary inclination of young adults towards plant-based protein sources, such as legumes and soy, reflects a significant generational shift in preferences and nutritional awareness. This trend can be attributed to a growing awareness of health, environmental sustainability and ethical concerns associated with the production of foods of animal origin. Younger demographics are increasingly embracing plant-based diets as part of a lifestyle that prioritises wellness and ecological responsibility [30]. Furthermore, the nutritional profile of legumes and soy, rich in fibre, vitamins and minerals while low in calories and saturated fat, is in line with the health values of this age group [31]. This change also reflects wider societal changes, where plant-based diets are no longer marginal lifestyle choices, but are becoming mainstream, driven by both health and environmental considerations [32]. The increasing availability and variety of plant-based foods has further facilitated this transition, making it easier for young adults to integrate these foods into their daily diet [33]. The results also suggest opportunities for targeted dietary interventions, such as promoting plant-based protein sources among the elderly, and nutrition education for young males on diversifying protein sources and reducing processed meat consumption [34]. A rather expected aspect emerging from our study is the implication of an omnivorous diet on the variety of protein sources and the consumption of specific foods, compared to other dietary regimes. Participants following the omnivorous diet consumed a broad spectrum of protein foods, which included meat, fish, dairy products, legumes and eggs. It is important to consider the extreme case of plant based diets that while presenting a potentially reduced choice of protein sources, mainly based on vegetables such as legumes, cereals, nuts and seeds, these diets may be well balanced and not necessarily associated with significant nutritional deficiencies. It is true that following a vegan diet may carry the risk of deficiencies in some micronutrients, such as vitamin B12, zinc, calcium and selenium. However, a low intake of micro- and macronutrients is rarely correlated with health impairment [35]. In addition to the variety of protein sources, the study found significant differences in the consumption of specific foods between diets. For example, in the omnivore diet, a higher consumption of red and processed meat was noted, while the pescatarian and vegetarian diets showed a higher consumption of foods such as legumes and soy products.

Our results highlight the significant impact of dietary choices on body composition (BMI), shedding light on the intricate relationship between dietary patterns, physical activity and health outcomes. Specifically, the vegetarian diet was associated with a lower BMI than other diets, echoing the results of the Adventist Health Study 2, which involved 96,000 subjects in the United States and Canada between 2002 and 2007 [36]. This lower BMI in vegetarians can be attributed to differences in energy absorption and utilisation, resulting in lower body mass and visceral adiposity. However, factors other than adiposity, potentially inherent in vegetarian diets, could also contribute to this result [37,38]. In our study, women following a flexitarian diet were predominantly in the normal weight range, suggesting that this diet could be particularly beneficial for weight management in women. Men following a flexitarian diet, on the other hand, showed a wider distribution between the normal-weight and overweight categories, indicating a different impact of this diet on male body composition [39]. Participants on the pescatarian diet, regardless of gender, fall mainly into the categories of normal weight and overweight, showing a similar trend to that of flexitarians. In contrast, the omnivorous diet group, especially among women, tends to shift towards the overweight and obesity categories. This observation raises questions about the potential calorie density and nutrient composition of omnivorous diets, which could contribute to a higher BMI. The variance in BMI distribution among omnivores also suggests that factors such as portion size, food quality and lifestyle choices, including physical activity levels, play a crucial role in determining overall body composition. These insights into how different dietary patterns influence body composition, particularly in relation to gender, emphasise the need for personalised nutritional recommendations. They also highlight the importance of considering individual lifestyle factors, such as exercise habits and food preferences, in developing effective strategies for weight management and optimising overall health. 

By exploring the interaction between dietary habits and physical activity, our study sheds light on how different combinations of diet and sport influence body composition. While previous research [40,41] does not suggest a significant impact of diet type on exercise capacity, our results indicate that diet choice, when combined with specific types of physical activity, can significantly influence body composition. Our analysis revealed that non-exercising vegetarians generally have a lower FM percentage (FM%) than their counterparts in other non-exercising diet groups. This suggests that, even without the inclusion of sport, certain dietary choices may inherently contribute to reducing FM. Interestingly, among athletic individuals, vegetarians participating in endurance sports and pescatarians engaged in strength sports showed a particularly low FM%, highlighting how specific combinations of diet and sport may optimise FM levels [42]. In contrast, a higher FM% was found in flexitarians engaged in endurance and team sports and among omnivores participating in endurance sports, emphasising the complex relationship between diet, type of sport and body composition. It appears that combining an omnivorous diet with team and strength activities, as well as a vegetarian diet with strength training, is favourable for improving lean mass levels, as supported by previous findings [43]. This is in line with our observation that, among athletes, strength training was more effective in increasing body protein in all food groups than resistance training. Furthermore, our study found that the ratio of FM/FFM was similar among the different food groups among non-athletes, with vegetarians having the lowest ratios. Among athletes, however, vegetarians participating in endurance sports showed the lowest FM/FFM ratios. In contrast, athletes from other dietary groups engaged in strength training showed lower FM/FFM ratios than athletes participating in endurance sports, suggesting that strength training, regardless of dietary patterns, may be more effective in improving this aspect of body composition. These results emphasise the importance of considering diet and physical activity together when optimising BC. Furthermore, our data suggest that further longitudinal studies evaluating specific combinations of dietary patterns and physical activities are needed.

This study has limitations. Firstly, the cross-sectional nature of the study design limits the ability to establish causality between dietary habits and health outcomes. Longitudinal studies would be needed to ascertain the long-term effects of these dietary patterns. Secondly, reliance on self-reported dietary data may introduce recall bias, potentially affecting the accuracy of the information collected. There is also an important difference in numerosity in the 4 groups with the vegetarian group much smaller than the others. Another limitation of our study is the absence of a formal validation test or pilot-scale trial for the questionnaire used. Although the questionnaire was designed in alignment with existing validated food taste questionnaires, the lack of a standalone validation process may affect the interpretation of our findings. This aspect should be considered when evaluating the conclusions drawn from our research. Another shortcoming of this study is the participant pool, which includes individuals specifically seeking expert assessment and guidance for their BC and related health concerns. While this provided valuable insights into a health-conscious subset, these individuals may not fully represent the broader general population, due to their distinct health-seeking behaviours and potential presence of specific health concerns. This factor should be considered when generalizing the findings to a wider audience. Finally, although the study included a heterogeneous group of participants, it was limited to a specific geographical area, which may limit the applicability of the results to other populations with different cultural and dietary habits.

## 5. Conclusions

In conclusion, our study highlights the significant influence of dietary patterns and physical activity on fat mass, emphasising their essential role in maintaining health and preventing chronic diseases. We identified distinct gender and age differences in food consumption, with younger males favouring meat and animal-based proteins and females of all ages leaning more towards plant-based proteins. These trends reflect a combination of personal health choices and broader cultural influences. In particular, our results show that vegetarian diets, especially when associated with endurance sports, are associated with lower fat mass percentages. This suggests that such diets, even in the absence of intense physical activity, may be effective in managing body fat. On the other hand, omnivorous and flexitarian diets, if not accompanied by adequate physical activity, tend to be linked to higher fat mass, indicating the need for a balanced approach to diet and lifestyle. This research contributes to the nuanced understanding of how diet and exercise intertwine to influence health outcomes, emphasising the need for personalised diet and lifestyle interventions. Although the study’s reliance on self-reported data and the demographic specificity of participants limit the breadth of their implication, these findings provide a basis for future research to explore these dynamics in more diverse and extended populations.

## Figures and Tables

**Figure 1 foods-13-00529-f001:**
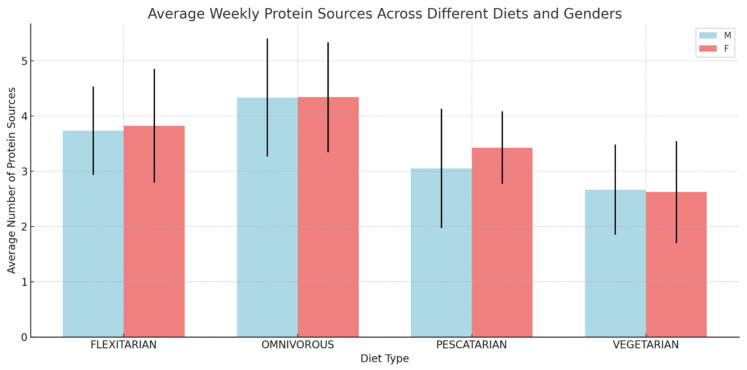
Dietary Patterns and Gender Disparity in Protein Source Diversity. The bar graph represents the mean number of different protein sources consumed on a weekly basis, segmented by diet type (Flexitarian, Omnivorous, Pescatarian, Vegetarian) and gender (males in light blue, females in light red). Error bars indicate standard deviation from the mean. Note: Data are presented as mean ± standard deviation. The dietary intake information was self-reported and categorized according to the predominant dietary patterns. ‘M’ denotes males and ‘F’ represents females. Independent *t*-tests were conducted to compare the mean number of protein sources between genders within each dietary category, resulting in non-significant *p*-values. An ANOVA was performed to compare the mean number of protein sources across the four dietary patterns, yielding a highly significant *p*-value (<0.001).

**Figure 2 foods-13-00529-f002:**
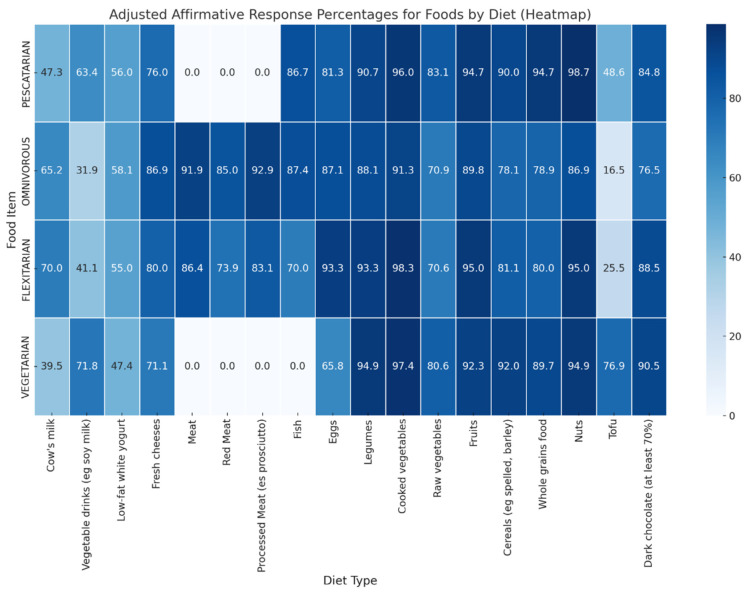
Consumption Patterns of Food Items Across Different Diets. This heatmap shows the percentage of affirmative (‘Yes’) responses for the consumption of various food items across four diet types. Data presented reflect the proportion of participants who reported consuming each food item, with percentages calculated based on affirmative responses. Dietary adjustments were applied prior to analysis: for Pescatarians and Vegetarians, meat and processed meat consumption was set to zero; for Vegetarians, fish consumption was also set to zero. Chi-squared tests for independence were performed to evaluate the significance of differences in consumption across diet types. Significant differences were found for cow’s milk (*p* < 0.001), vegetable drinks (*p* < 0.001), fresh cheeses (*p* < 0.001), meat (*p* < 0.001), red meat (*p* < 0.001), processed meat (*p* < 0.001), fish (*p* < 0.001), eggs (*p* < 0.001) and tofu (*p* < 0.001). No significant differences were observed for low-fat white yogurt (*p* = 0.104), legumes (*p* = 0.855), cooked vegetables (*p* = 0.482), raw vegetables (*p* = 0.290), fruits (*p* = 0.826), cereals (*p* = 0.557), whole grains (*p* = 0.211), nuts (*p* = 0.228) and dark chocolate (*p* = 0.770).

**Figure 3 foods-13-00529-f003:**
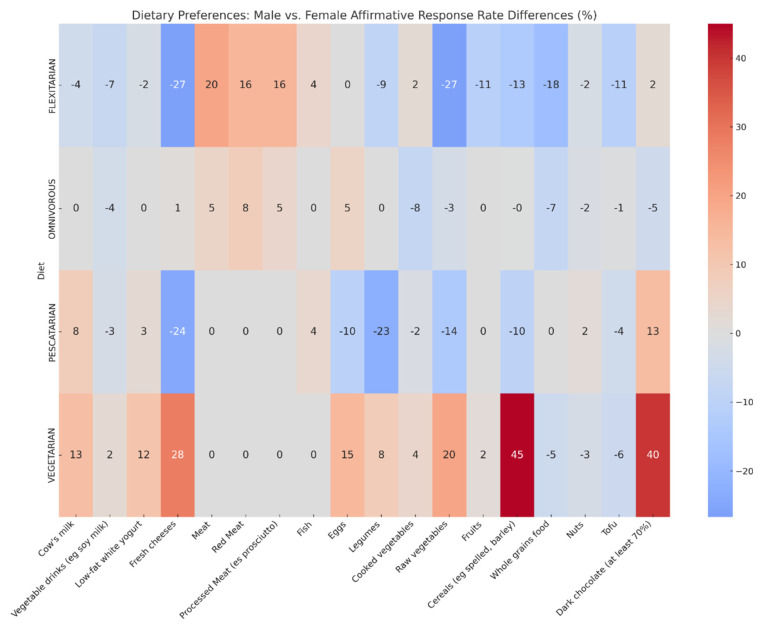
Dietary Preferences by Gender: A Comparative Analysis Across Diets. This heatmap illustrates the percentage-point differences in affirmative response rates to various food items between males and females across four dietary patterns: Flexitarian, Omnivorous, Pescatarian, and Vegetarian. Statistical significance was assessed using two-proportion z-tests for each food item across diets. *p* values: Cow’s milk: 0.95; Vegetable drinks (e.g., soy milk): 0.126; Low-fat white yogurt: *p* = 0.951; Fresh cheeses: *p* = 0.91; Meat: *p* = 0.0002; Red Meat: *p* = 0.002; Processed Meat (e.g., prosciutto): *p* = 0.0004; Fish: *p* = 0.562; Eggs: *p* = 0.009 Legumes: *p* = 0.921; Cooked vegetables: *p* < 0.00001; Raw vegetables: *p* = 0.1326; Fruits: *p* = 0.9423; Cereals (e.g., spelt, barley): *p* = 0.8124 Whole grains: *p* = 0.0009; Nuts: *p* = 0.326; Tofu: *p* = 0.58; Dark chocolate (at least 70%): *p* = 0.1233.

**Figure 4 foods-13-00529-f004:**
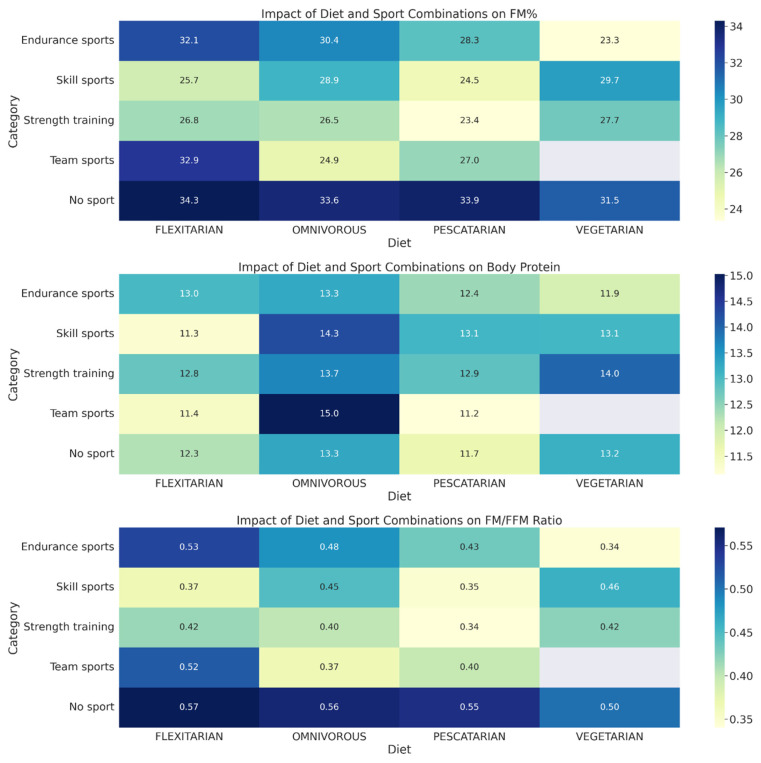
Associations Between Dietary Patterns, Athletic Engagement, and Body Composition Metrics. This heatmap illustrates the impact of various diets and sports activities on body composition, specifically focusing on averages of FM percentage (FM%), body protein content, and FM/FFM ratio. Differences across dietary (Flexitarian, Omnivorous, Pescatarian, Vegetarian) and sport (endurance, skill, strength training, team, no sport) categories are highlighted, with colour gradients representing the magnitude of each metric. The normality of data was tested with the Shapiro–Wilk test, revealing a mix of normal and non-normal distributions across different groupings. Statistical significance was assessed using the Kruskal–Wallis test. Significant differences across groups were found—FM% (*p* < 0.001), FM/FFM Ratio (*p* < 0.001) and Body Protein (*p* = 0.0016).

**Table 1 foods-13-00529-t001:** Demographic and Anthropometric Analysis of Patients by Gender and Age Group. Distribution of patients by gender and age group with anthropometric details and comparative assessments between males and females.

		Total (*n*)	M	F	*p*-Value
Total	*n*	1342	553 (41.2%)	789 (58.8%)	<0.0001
Age	yrs	39.4 ± 11.8	38.2 ± 11.6	40.2 ± 11.8	<0.0001
18–30	*n* (%)	326 (24.3%)	146 (44.8%)	180 (55.2%)	0.007
31–45		566 (42.2%)	248 (43.8%)	318 (56.2%)	
46–65		450 (33.6%)	159 (35.3%)	291 (64.7%)	
Weight	kg	79.9 ± 17.6	89.5 ± 17.2	73.3 ± 14.6	<0.0001
BMI	kg/m^2^	28.1 ± 5.2	28.8 ± 5.1	27.6 ± 5.2	<0.0001
<18.5	*n* (%)	9 (0.7%)	2 (22.2%)	7 (77.8%)	0.005
18.5–24.9		391 (31.2%)	131 (33.5%)	260 (66.5%)	
25.0–29.9		531 (42.3%)	225 (42.4%)	306 (57.6%)	
30.0–34.9		275 (21.9%)	133 (48.4%)	142 (51.6%)	
35.0–39.9		104 (8.3%)	48 (46.2%)	56 (53.8%)	
40.0–44.9		24 (1.9%)	10 (41.7%)	14 (58.3%)	
≥45		8 (0.6%)	4 (50.0%)	4 (50.0%)	
FM	kg	25.0 ± 10.7	23.2 ± 10.9	26.2 ± 10.4	<0.0001
FM	%	30.6 ± 9.0	24.8 ± 7.6	34.6 ± 7.5	<0.0001
AC	cm	96.8 ± 14	101.1 ± 14.1	93.8 ± 13.1	<0.0001
FFM	kg	52.2 ± 11.2	62.9 ± 8.1	44.6 ± 5.3	<0.0001
Body Water	lt	38.8 ± 8.4	46.7 ± 6.1	33.2 ± 4.4	<0.0001
BMR	Kcal	1656.1 ± 342.1	1963.5 ± 272.6	1439.8 ± 183.9	<0.0001

Data are expressed as mean ± standard deviation for continuous variables and as count (percentage) for categorical variables. The *p*-values are calculated by Student’s *t*-test for continuous variables and Chi-square test of independence for categorical variables to assess differences between males and females. A *p*-value of less than 0.05 is considered statistically significant. ‘BMI’ refers to Body Mass Index, ‘FM’ to Fat Mass, ‘AC’ to Abdominal Circumference, ‘FFM’ to Lean Mass, ‘BMR’ to Basal Metabolism Rate. Age groups are defined as follows: 18–30 years, 31–45 years, 46–65 years. The BMI classes are divided according to World Health Organisation standards.

**Table 2 foods-13-00529-t002:** BMI Distribution Across Different Diets by Gender. This table displays the distribution of body mass index (BMI) categories across various diets (Flexitarian, Omnivorous, Pescatarian, Vegetarian) by gender (Female, Male). Each cell shows the number of individuals and their corresponding percentage within the specific BMI category for each diet and gender group.

		BMI						
Diet	(*n*)	<18.5	18.5–24.9	25.0–29.9	30.0–34.9	35.0–39.9	40.0–44.9	≥45
Flexitarian	F (42)	0 (0.0%)	18 (42.9%)	13 (31.0%)	5 (11.9%)	5 (11.9%)	1 (2.4%)	0 (0.0%)
	M (13)	0 (0.0%)	6 (46.2%)	4 (30.8%)	3 (23.1%)	0 (0.0%)	0 (0.0%)	0 (0.0%)
	Total (55)	0 (20.0%)	24 (43.6%)	17 (30.9%)	8 (14.5%)	5 (9.1%)	1 (1.8%)	0 (0.0%)
Omnivorous	F (670)	5 (0.7%)	210 (31.3%)	263 (39.3%)	125 (18.7%)	50 (7.5%)	13 (1.9%)	4 (0.6%)
	M (507)	2 (0.4%)	112 (22.1%)	207 (40.8%)	126 (24.9%)	47 (9.3%)	9 (1.8%)	4 (0.8%)
	Total (1177)	7 (0.6%)	322 (27.4%)	470 (39.9%)	251 (21.3%)	97 (8.2%)	22 (1.9%)	8 (0.7%)
Pescatarian	F (53)	0 (0.0%)	27 (50.9%)	17 (32.1%)	9 (17.0%)	0 (0.0%)	0 (0.0%)	0 (0.0%)
	M (19)	0 (0.0%)	10 (52.6%)	8 (42.1%)	0 (0.0%)	1 (5.3%)	0 (0.0%)	0 (0.0%)
	Total (72)	0 (25.0%)	37 (51.4%)	25 (34.7%)	9 (12.5%)	1 (1.4%)	0 (25.0%)	0 (0.0%)
Vegetarian	F (24)	2 (8.3%)	5 (20.8%)	13 (54.2%)	3 (12.5%)	1 (4.2%)	0 (0.0%)	0 (0.0%)
	M (14)	0 (0.0%)	3 (21.4%)	6 (42.9%)	4 (28.6%)	0 (0.0%)	1 (7.1%)	0 (0.0%)
	Total (38)	2 (5.3%)	8 (21.1%)	19 (16.7%)	7 (18.4%)	1 (2.6%)	1 (2.6%)	0 (0.0%)

(*n*) indicates the total number of individuals in each dietary category for each gender. Percentages are calculated based on the total number in each diet-gender group. BMI categories are divided as follows: <18.5 (Underweight), 18.5–24.9 (Normal weight), 25.0–29.9 (Overweight), 30.0–34.9 (Obese Class I), 35.0–39.9 (Obese Class II), 40.0–44.9 (Obese Class III), and ≥45 (Extremely Obese). Data is rounded to one decimal point. Statistical Analysis Note: The Chi-square test of independence was applied to examine the relationships between diet types, gender and BMI categories. For the relationship between diet types and BMI categories, the Chi-square statistic was 45.61 with a *p*-value of 0.00034, suggesting a significant association between the type of diet and BMI distribution. For the relationship between gender and BMI categories, the Chi-square statistic was 18.34 with a *p*-value of 0.00545, indicating a significant association between gender and BMI distribution. These tests suggest that both diet type and gender are statistically associated with variations in BMI categories. However, it is important to note that these tests do not imply causation but only indicate an association. The expected frequencies in the contingency tables were calculated based on the overall distribution of the sample. Degrees of freedom for the diet-BMI and gender-BMI tests were 18 and 6, respectively. *p*-values less than 0.05 were considered statistically significant.

**Table 3 foods-13-00529-t003:** Comparative Analysis of Dietary Protein Sources Consumption Across Different Age Groups and Genders. Table representing the percentage of population consuming various protein sources above the 75th percentile, categorised by age group and gender. Letters indicate significant differences in consumption between males and females within each age group.

	Total				M				F			
Age Group	18–30	31–45	46–65	*p*-Value	18–30	31–45	46–65	*p*-Value	18–30	31–45	46–65	*p*-Value
Meat	28.8%	24.6%	22.6%	0.094	42.8% a	30.4% a	24.1% b	**0.004**	17.4% a	20.0% a	21.8% b	0.616
Processed Meat	18%	17.8%	18.3%	0.925	26.2% a	25.5% a	24.7% a	0.808	11.2% a	11.8% a	14.9% a	0.143
Fish	21.7%	20.3%	19.9%	0.823	29% a	25.9% a	30.4% a	0.817	15.7% a	15.9% a	14.2% a	0.431
Eggs	14.9%	10.1%	8.5%	0.112	21.4% a	12.2% b	14.6% a	0.45	9.6% a	8.6% b	5.2% a	0.098
Dairy	14.2%	15.3%	17.9%	0.184	20% a	17.4% b	20.3% b	0.28	9.6% a	13.7% b	16.6% b	0.417
Legumes	23.8%	21.9%	21.9%	0.787	22.1% b	25.1% b	23.4% b	0.641	25.3% b	19.4% b	21.1% b	0.556
Soy	12.1%	12.8%	5.4%	**0.002**	8.3% b	10.9% b	3.2% b	**0.019**	15.2% b	14.3% b	6.6% b	**0.024**
n. protein sources	11.2%	11.9%	8.8%	0.213	11.5% b	17.8% b	14% a	0.876	11% b	19.4% b	7.5% a	0.673

“a” indicates a statistically significant difference (*p*-value < 0.05); “b” indicates no significant difference (*p*-value ≥ 0.05). *p*-values for comparison between males and females in each age group: Meat: 18–30 (3.09 × 10^6^), 31–45 (0.006), 46–65 (0.566). Processed Meat: 18–30 (0.00033), 31–45 (0.00024), 46–65 (0.0054). Fish: 18–30 (0.0055), 31–45 (0.0011), 46–65 (0.0009). Eggs: 18–30 (0.0034), 31–45 (0.144), 46–65 (0.0036). Dairy: 18–30 (0.0054), 31–45 (0.696), 46–65 (0.205). Legumes: 18–30 (0.787), 31–45 (0.190), 46–65 (0.860). Soy: 18–30 (0.079), 31–45 (0.262), 46–65 (0.340). Number of Protein Sources: 18–30 (0.761), 31–45 (0.866), 46–65 (0.0033).

## Data Availability

The datasets produced and analyzed during the present study are obtainable from the corresponding author upon reasonable request.

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
