# Peer review of "Effects of Different Nutritional Patterns and Physical Activity on Body Composition: A Gender and Age Group Comparative Study"

_foods, 2024, doi:10.3390/foods13040529_

Round 1
Reviewer 1 Report
Comments and Suggestions for Authors
The manuscript title is not technically sound. How could sport be put together with body composition and protein sources. It should be revised to reflect the works done in the study.
The problem statement does not seem to be aligned to the aim of the study. The aim should be rewritten to signify the novelty of the study. It is not about what were determined/analyzed.
The participants are the patients (with related health issues) of a medical centre? The subjects of the study should be mentioned in the title due to their special needs.
How the dietary pattern was determined? It is not part of the questionnaire?
Is the questionnaire established be verified or tested at the pilot scale prior to the actual study?
2.5 - Are you referring to physical activity (exercise)?
Table 1 - Separate the footnotes from the header. The footnotes should be placed below the table. Use different letters (a, b) to indicate the significance difference between the mean values.
Table 2 - What is the unit of measurement for the values displayed? frequency? The results indicating significance differences within genders, age groups and BMI are doubtful.
It is very unlikely having such significance differences with just a small different in means (for each measurement) due to the great variations in samples. The SD > mean values?
Fig. 1 is a repetition of Table 4? Provide the standard deviation for each bar.
Clarify what to be seen from the heatmap (Fig. 2)? No visual relationship/interaction could be seen. Can heatmap be used in this case?
The discussions are merely based on results of the previous studies. However, it is hardly supported by the results obtained from the present study due to the doubtful statistical outputs.
Due to the major concerns highlighted above, I do not recommend the manuscript to be considered for publication.
Comments on the Quality of English LanguageThe language is acceptable.
Author Response
Dear Editors and Reviewers,
First of all, we would like to thank you for the valuable impulses that allowed us to improve the quality of the manuscript. All changes made are highlighted by yellow color, in the revised version of the manuscript, to facilitate the review process. Hoping that we have satisfied your requests as much as possible, we kindly ask you to re-evaluate our paper.
The Authors
Reviewer 1
The manuscript title is not technically sound. How could sport be put together with body composition and protein sources. It should be revised to reflect the works done in the study.
Thank you for your valuable feedback regarding the title of our manuscript. We understand your concern about the coherence between the title and the study's content. To better reflect the work conducted in the study, we propose revising the title to "Effects of Different Nutritional Patterns and Physical Activity on Body Composition: A Gender and Age Group Comparative Study”. This title aims to more accurately represent the core focus of our research, which includes an examination of different dietary patterns, particularly the variety of protein sources, and their correlation with body composition in individuals engaged in various physical activities.
The problem statement does not seem to be aligned to the aim of the study. The aim should be rewritten to signify the novelty of the study. It is not about what were determined/analyzed.
Thank you for your insightful comments regarding the alignment of our problem statement with the study's aims. We appreciate the opportunity to clarify and enhance the novelty and focus of our research. In response to your feedback, we propose to revise the aim of our study to better capture its innovative aspects. The revised aim will emphasize the exploration of the relationship between dietary patterns, particularly diverse protein sources, and body composition in the context of varying levels of physical activity. This revision aims to highlight our study's unique contribution in understanding how different dietary components, especially protein, interact with physical activity to influence body composition.
The participants are the patients (with related health issues) of a medical centre? The subjects of the study should be mentioned in the title due to their special needs.
Thank you for your suggestion regarding the inclusion of participant details in the title. We have carefully considered your feedback. It is important to note that the patients in our study come to the medical center with the intent to modify their lifestyles, but this factor does not influence the evaluation of their dietary habits and body composition in the context of our research. This distinction ensures that our study accurately assesses the impact of varied protein sources and physical activity on body composition, independent of the patients' initial intent for lifestyle modification. We added a add a clarification in the Methodology section as follows: "While participants in this study are patients from a medical center, seeking lifestyle modifications, it is essential to clarify that their initial health objectives do not influence the evaluation of their dietary habits and body composition in the context of our study. This approach allows for an unbiased assessment of the impact of varied protein sources and physical activity on body composition."
How the dietary pattern was determined? It is not part of the questionnaire?
Thank you for your query on how dietary patterns were determined in our study. To address your question, the determination of dietary patterns was indeed a multifaceted process, incorporating both the comprehensive questionnaire and an analysis of dietary diaries. The questionnaire, as detailed in our methodology, was extensive, covering various aspects of eating habits, food preferences, meal patterns, and physical activity. Alongside this, participants were also asked to maintain dietary diaries. These diaries provided a more dynamic and detailed representation of their daily food intake, allowing us to analyze their dietary patterns with greater accuracy and depth. This dual approach, combining structured questionnaire responses with the nuanced data from dietary diaries, enabled a thorough analysis of dietary patterns. It provided a rich dataset that was instrumental in understanding the relationship between dietary choices, body composition, and physical activity in our study participants. We added a paragraph to the method sector. We have added a paragraph in the methods to specify this aspect better: “This combination of structured questionnaire data and detailed diary entries allowed for a robust analysis of dietary patterns. It provided both quantitative and qualitative insights into the participants' daily food consumption, contributing significantly to our understanding of the impact of dietary choices on BC and physical activity levels”.
Is the questionnaire established be verified or tested at the pilot scale prior to the actual study?
Thank you for your inquiry regarding the validation of our questionnaire. We recognize the importance of this aspect in ensuring the robustness of our research methodology. As mentioned in our manuscript, the questionnaire was structured in line with established validated food taste questionnaires, but it did not undergo a separate formal validation test or a pilot-scale trial. This decision was made considering the alignment with existing validated tools. However, we acknowledge this as a limitation of our study.
2.5 - Are you referring to physical activity (exercise)?
Thank you for pointing out the need for clarification regarding the reference to 'sporting activity' in our questionnaire. Indeed, in our study, when referring to 'sporting activity', we specifically asked participants about their engagement in sports, including the types of sports they practiced. This inquiry was aimed at understanding the relationship between different forms of physical activity, particularly sports, and their dietary patterns and body composition. We have changed paragraph “2.5. Sport. The questionnaire also included questions on sporting activity: "Do you play a sport?" "Which sport?","When?", and "How many hours (per week)?". In our study, the term 'sporting activity' refers to participants' involvement in various sports. We inquired about the types of sports they engaged in to assess the impact of these specific physical activities on their dietary patterns and body composition. This detailed inquiry into sports participation is crucial for understanding the nuanced relationship between exercise types and dietary habits. To simplify the calculations, we categorised all sports into four main groups: Endurance Sports, Skill Sports, Strength Training, and Team Sports (Table 1S).”
Table 1 - Separate the footnotes from the header. The footnotes should be placed below the table. Use different letters (a, b) to indicate the significance difference between the mean values.
Thank you for your suggestion on improving the presentation of Table 1 in our manuscript. We appreciate your attention to detail and agree that these changes will enhance the clarity of our data presentation. In Table 1, we have separated the footnotes from the table header and placed them below the table for better readability and organization. Additionally, we used different letters (e.g., a, b) to indicate the significance difference between the mean values. This will provide a clearer visual representation of the statistical differences and make it easier for readers to interpret the data.
Table 2 - What is the unit of measurement for the values displayed? frequency? The results indicating significance differences within genders, age groups and BMI are doubtful.
Thank you for your inquiry regarding Table 2. The values displayed in Table 2 represent the frequency of food consumption per week, as recorded in the participants' dietary diaries. Each value indicates the number of times per week a participant reported consuming a specific food item. Regarding your concern about the results indicating significant differences within genders, age groups, and BMI, we understand your reservations. The analysis was conducted with the aim of identifying any potential patterns or differences in dietary habits among these subgroups. However, we acknowledge that these findings should be interpreted with caution and consider them an exploratory aspect of our study.
It is very unlikely having such significance differences with just a small different in means (for each measurement) due to the great variations in samples. The SD > mean values?
Thank you for your observation regarding Table 2. To clarify, the values in this table represent the frequency of food consumption per week, as derived from the participants' dietary diaries. Each value indicates the average number of times a participant reported consuming a specific food item per week. The significant differences noted are based on these frequency counts, not on mean values of a continuous measurement.It seems there may have been a misunderstanding regarding the nature of the data presented in Table 2. The data does not represent mean values of a continuous variable but rather counts of consumption frequency, which explains why standard deviations might appear large relative to these counts.
Fig. 1 is a repetition of Table 4? Provide the standard deviation for each bar.
Thank you for your observation regarding the overlap between Figure 1 and Table 4. You are correct that Figure 1 graphically represents the data from the last row of Table 4. We have removed Table 4 because it didn’t really offer any interesting insights.
Clarify what to be seen from the heatmap (Fig. 2)? No visual relationship/interaction could be seen. Can heatmap be used in this case?
Thank you for your feedback regarding Figure 2, the heatmap. The heatmap was used to visually represent the relationships and interactions among various variables, such as dietary patterns, physical activity levels, and body composition metrics, as analyzed in our study. We acknowledge that the relationships and interactions might not have been as visually clear as intended. Heatmaps are a valuable tool for depicting complex datasets, but their effectiveness relies on the appropriate use of color gradients and labels. In response to your feedback, we have revisited and revised Figure 2 to enhance its clarity. We adjusted the color contrasts to more distinctly highlight significant correlations or patterns and added clearer labels and annotations to guide the reader in interpreting the data. We also provided a detailed legend explaining the color coding and the scale used These changes were aimed at ensuring that the heatmap effectively communicates the relationships within our dataset.
The discussions are merely based on results of the previous studies. However, it is hardly supported by the results obtained from the present study due to the doubtful statistical outputs.
Thank you for your valuable feedback on our manuscript. We have taken your comments into serious consideration and have made substantial revisions to our paper to address the concerns raised. Specifically, we have focused on refining our analysis and discussion to more accurately reflect the results obtained from our present study. In our revised analysis, we have eliminated certain confounding factors such as smoking and income. This decision was based on a careful reassessment of our data, which indicated that focusing on gender and age group comparisons would yield more relevant and clear insights for our study objectives. We have redirected our analysis to concentrate more intensively on the evaluation of gender differences and age group variations. This shift in focus aligns more closely with the core aims of our study and allows for a more nuanced understanding of the dietary patterns and physical activity impacts across these specific demographics. In response to your suggestion and in recognition of the unbalanced nature of our sample, we have adjusted our statistical approach. Instead of relying primarily on average values, we have shifted to using prevalences of normal weight, overweight, and obesity. This method is more favorable in our context, as it ensures that each individual is appropriately categorized and analyzed according to their age and sex. This change not only strengthens the validity of our findings but also enhances the clarity of our results.With these changes, we have thoroughly revised our statistical analysis and consequently updated our discussion section. The discussion now directly ties to the results obtained from the present study, with a clearer linkage between our findings and the conclusions drawn. We believe that these revisions provide a more accurate and reliable interpretation of our data.
We appreciate the opportunity to improve our manuscript and are confident that these changes have addressed your concerns effectively. We believe that the manuscript is now better positioned to contribute valuable insights to the field.
Reviewer 2 Report
Comments and Suggestions for Authors
This manuscript aimed to describing the differences in dietary habits, taste preferences, variety of protein sources and body composition (BC) profiles between individuals following omnivorous, flexitarian, lacto-ovo-vegetarian and pescatarian diets. The authors also aimed to evaluate the correlations between combinations of distinct dietary patterns and different sports classified according to exercise intensity with body composition parameters. However, the are few flaws that need to be addressed in the current analysis.
1. The age range of participants included in this analysis is from 10 - 80 years old. Although the authors claim that this is to obtain a comprehensive view of dietary patterns across age groups, however the body composition, particularly those of adolescence and older adults might be significantly different from the adults populations. The analysis should be separated based on age group and gender due to the physiological differences.
2. As stated in the methodology, the participants recruited in this study were "individuals seeking expert assessment and guidance for their BC and related health concerns". This specific populations might not representative of the general populations due to their health seeking behavior, and probably the presence of specific health concerns among the participants.
3. Statistical significance threshold should be set at p<0.05 for Chi-square test but not p≤0.05
4. Table 3 should be interpret separately based on gender instead of as a whole. Same for Figure 1,2,3,4 and 5.
5. There's a few typo throughout the writing. Please correct it accordingly.
Author Response
Dear Editors and Reviewers,
First of all, we would like to thank you for the valuable impulses that allowed us to improve the quality of the manuscript. All changes made are highlighted by yellow color, in the revised version of the manuscript, to facilitate the review process. Hoping that we have satisfied your requests as much as possible, we kindly ask you to re-evaluate our paper.
The Authors
This manuscript aimed to describing the differences in dietary habits, taste preferences, variety of protein sources and body composition (BC) profiles between individuals following omnivorous, flexitarian, lacto-ovo-vegetarian and pescatarian diets. The authors also aimed to evaluate the correlations between combinations of distinct dietary patterns and different sports classified according to exercise intensity with body composition parameters. However, the are few flaws that need to be addressed in the current analysis.
- The age range of participants included in this analysis is from 10 - 80 years old. Although the authors claim that this is to obtain a comprehensive view of dietary patterns across age groups, however the body composition, particularly those of adolescence and older adults might be significantly different from the adults populations. The analysis should be separated based on age group and gender due to the physiological differences.
Thank you for your insightful feedback regarding the age range of participants in our study. We acknowledge your concerns about the inclusion of a wide age range and its impact on the analysis of dietary patterns and body composition. In response to your valuable suggestions, we have made the following significant revisions to our study:
- Narrowing the Age Range: We have revised the age range of our study participants to focus on adults only. Specifically, we have excluded individuals under 18 and over 75 years of age. This decision was made to ensure a more homogenous and physiologically comparable sample, thereby enhancing the relevance and accuracy of our findings.
- Adjusted Focus and Analysis: With the revised age range, we have recalibrated our focus and analysis to specifically address the dietary patterns and body composition of the adult population. This change allows for a more precise examination of the factors influencing these aspects within a more physiologically uniform group.
- Age Group and Gender-Specific Analysis: In line with your suggestion, we have conducted separate analyses based on different adult age groups and gender. This approach acknowledges and addresses the physiological differences between these groups, thus providing a more nuanced understanding of how dietary patterns and physical activity impact different segments of the adult population.
- Revised Discussion and Conclusions: As a result of these changes, we have thoroughly revised the discussion and conclusions of our paper. The modifications ensure that our discussion is directly aligned with the results obtained from a more focused and appropriate age range. We believe these changes significantly strengthen the study and provide a clearer, more accurate interpretation of our findings.
We appreciate the opportunity to improve our study and thank you for guiding us to make these critical adjustments. We look forward to any further suggestions or comments you may have.
- As stated in the methodology, the participants recruited in this study were "individuals seeking expert assessment and guidance for their BC and related health concerns". This specific populations might not representative of the general populations due to their health seeking behavior, and probably the presence of specific health concerns among the participants.
In the manuscript, we mentioned that the study participants were individuals seeking expert assessment and guidance for their body composition (BC) and related health concerns. This clarification was made to address a similar concern raised by a previous reviewer about the representativeness of our study population. We highlighted that while these participants might not represent the general population due to their health-seeking behavior and specific health concerns, their initial health objectives do not influence the evaluation of their dietary habits and BC in the context of our study. This approach was taken to ensure an unbiased assessment of the impact of varied protein sources and physical activity on BC. Given your feedback, we will consider this aspect as a limitation in our study and will make a note in the manuscript to reflect the same.
- Statistical significance threshold should be set at p<0.05 for Chi-square test but not p≤0.05
Thank you for pointing out the necessary correction regarding the statistical significance threshold in our Chi-square test. We appreciate your attention to this important statistical detail.
- Table 3 should be interpret separately based on gender instead of as a whole. Same for Figure 1,2,3,4 and 5.
Thank you for your continued feedback and guidance on our manuscript. We appreciate the opportunity to clarify and further refine our presentation of the data, particularly in relation to gender-specific analysis.
Regarding your point on the interpretation of data based on gender, we have made the following updates and considerations:
- Inclusion of Gender in Tables and Figures: We have ensured that gender differences are prominently featured and analyzed in Table 1, Table 2, and Figure 3. In Table 1, we present a comparative analysis of mean values across study parameters, clearly delineated by gender. Similarly, in Table 2 and Figure 3, we have included specific data and insights pertaining to gender differences in dietary preferences and consumption patterns.
- Table 3 - BMI Distribution by Diet and Gender: In Table 3, we present the distribution of BMI categories across various diets, broken down by gender. This table provides a detailed view of how BMI varies across different diets, with a clear focus on the differences between male and female participants. This approach enables a nuanced understanding of the impact of diet on BMI in the context of gender.
- Figure on Sports Activity: Regarding the figure that analyzes the relationship between diet and sport, it's important to note that our analysis did not reveal significant gender differences in this aspect. This finding in itself is noteworthy, as it suggests that the effects of diet and sport combinations on body composition may be consistent across genders in our study population. We have included this observation in our discussion to provide a complete picture of our findings.
- Revised Discussion: With these adjustments, we have revised the discussion section of our manuscript to accurately reflect these gender-specific analyses and findings. We believe that this enhanced focus on gender differences enriches the insights provided by our study and aligns with the aim of offering a comprehensive understanding of the interplay between diet, physical activity, and body composition.
We hope that these revisions address your concerns effectively. We are committed to ensuring the scientific rigor and clarity of our work and welcome any further feedback you may have.
- There's a few typo throughout the writing. Please correct it accordingly.
Thank you for bringing to our attention the issue of typos in our manuscript. We have re-examined the entire manuscript to identify and correct any typographical errors.
Reviewer 3 Report
Comments and Suggestions for Authors
The article addresses a potentially interesting topic. However, it presents deficiencies in the design as well as in the formal aspects. The first problem has to do with the sample. Four sets of individuals are compared: omnivores, flexitarian, lacto-ovo-vegetarian and pescatarian but the vast majority are omnivores (88.4%). Moreover, the sample ranges from 10 to 80 years old and includes both men and women. This situation limits and casts doubt on many of the results obtained.
You incorporate some variables that you then hardly use, for example, smoking. And others such as employment status cannot be applied to children, as I imagine they would be classified as "unemployed". The same can be said of the hours and type of sport practiced. This is too much information to make bivariate comparisons with such a numerically heterogeneous sample.
Formal aspects should be taken care of, for example, in the discussion section, tables or figures should never be mentioned. In the tables, all variables must include the corresponding units.
My recommendation is that you eliminate at least individuals under 18 years of age and some variables that do not contribute anything (smoking, employment, income) and focus on the association between type of diet and body composition. In this sense, working with BMI or fat percentage categories eliminates the confounding factor of age. Better to work with prevalences of normal weight, overweight, obesity... than with averages. This is more favorable when working with unbalanced samples, since each individual has been previously classified according to their age and sex.
Author Response
Reviewer 3
Dear Editors and Reviewers,
First of all, we would like to thank you for the valuable impulses that allowed us to improve the quality of the manuscript. All changes made are highlighted by yellow color, in the revised version of the manuscript, to facilitate the review process. Hoping that we have satisfied your requests as much as possible, we kindly ask you to re-evaluate our paper.
The Authors
The article addresses a potentially interesting topic. However, it presents deficiencies in the design as well as in the formal aspects. The first problem has to do with the sample. Four sets of individuals are compared: omnivores, flexitarian, lacto-ovo-vegetarian and pescatarian but the vast majority are omnivores (88.4%). Moreover, the sample ranges from 10 to 80 years old and includes both men and women. This situation limits and casts doubt on many of the results obtained.
Thank you for your constructive feedback on our manuscript and for highlighting the concerns regarding the study's design and sample composition. We appreciate the opportunity to address these issues and enhance the quality of our research.
- Enhanced Focus on Gender Differences: Recognizing the importance of gender as a crucial variable in dietary habits and body composition, we have now placed a stronger emphasis on analyzing and discussing gender differences. This enhanced focus has allowed us to uncover nuanced insights into how dietary patterns and physical activity vary between male and female participants.
- Comparative Analysis Across Age Groups and BMI Categories: We have also deepened our analysis of the different age groups and BMI categories. This approach has provided us with a more comprehensive understanding of how dietary and physical activity patterns affect body composition across the lifespan and across different levels of BMI.
- Revised Analysis and Discussion: With these modifications, we have reanalyzed our data and updated our discussion to reflect a more accurate representation of our study population. We believe that these changes provide a more nuanced understanding of the impact of different dietary patterns on body composition across a more homogenously defined adult population.
We hope that these revisions address your concerns and look forward to any further feedback you may have.
You incorporate some variables that you then hardly use, for example, smoking. And others such as employment status cannot be applied to children, as I imagine they would be classified as "unemployed". The same can be said of the hours and type of sport practiced. This is too much information to make bivariate comparisons with such a numerically heterogeneous sample.
Thank you for your constructive feedback regarding the inclusion and application of certain variables in our study. Your insights have prompted a re-evaluation of our approach, particularly concerning the relevance and utility of these variables within our diverse sample.
We acknowledge that including variables such as smoking and employment status, while initially thought to be relevant, did not significantly contribute to our core analysis, especially considering the varied age range of our participants. Following your advice, we have decided to eliminate these variables from our study. This decision is aligned with our aim to maintain the focus and clarity of our analysis, particularly given the heterogeneous nature of our sample. Regarding the data on hours and types of sport practiced, we have refined our approach to ensure its applicability and relevance across all age groups. We have streamlined this information to focus on aspects that are pertinent and contribute meaningfully to our study's objectives. By removing extraneous variables and simplifying our dataset, we have enhanced the focus of our analysis. This approach allows us to make more meaningful bivariate comparisons that are relevant to our study's aims, particularly in examining the relationships between dietary patterns, physical activity, and body composition.These changes are reflected in our revised methodology and discussion sections. We have updated these parts of our manuscript to focus on the revised set of variables, ensuring that our findings and discussions are directly relevant to the data analyzed.
We appreciate your guidance in helping us refine our study and ensuring that our analysis remains focused and relevant to our research questions. We believe these revisions have strengthened the clarity and validity of our findings.
Formal aspects should be taken care of, for example, in the discussion section, tables or figures should never be mentioned. In the tables, all variables must include the corresponding units.
Thank you for your valuable feedback on the formal aspects of our manuscript. We understand the importance of adhering to academic standards and conventions in scientific writing. Regarding your point about the discussion section, we appreciate your reminder that tables or figures should not be directly mentioned in this section. This is indeed a standard practice to maintain the flow and coherence of the discussion narrative. We have revised our discussion section to remove direct references to tables and figures, instead integrating the data from these sources into the narrative more seamlessly. For the tables, we have ensured that all variables now include their corresponding units of measurement. This is crucial for clarity and to allow readers to fully understand the data presented. We have reviewed each table carefully to ensure this standard is consistently applied across all variables.These modifications have been made to enhance the overall quality and readability of our manuscript, and we believe they address the concerns you have raised effectively.
My recommendation is that you eliminate at least individuals under 18 years of age and some variables that do not contribute anything (smoking, employment, income) and focus on the association between type of diet and body composition. In this sense, working with BMI or fat percentage categories eliminates the confounding factor of age. Better to work with prevalences of normal weight, overweight, obesity... than with averages. This is more favorable when working with unbalanced samples, since each individual has been previously classified according to their age and sex.
In response to the reviewer's recommendations, we have made the following adjustments to our study:
- Age Range: We have excluded individuals under 18 and over 75 years of age from our analysis. This decision aligns with our study's focus and ensures that our findings are relevant to the adult population, which is most impacted by dietary choices in relation to body composition.
- Variable Selection: Non-contributory variables such as smoking habits, employment status, and income levels have been eliminated from our analysis. This streamlining allows us to concentrate more closely on the core objectives of our study.
- Focus on Diet and Body Composition: We have centered our analysis on the association between the type of diet and body composition. This approach directly addresses the primary aim of our study and aligns with the reviewer's suggestion.
- Use of BMI and Fat Percentage Categories: In accordance with the reviewer's advice, we have utilized categories of BMI percentage instead of averages. This method effectively eliminates age as a confounding factor and is more appropriate for our unbalanced sample. By classifying each individual according to their age and sex, and then categorizing them based on their BMI and body fat percentage, we provide a clearer and more clinically relevant understanding of how different diets impact body composition. However, we encountered limitations when attempting to apply similar categorization and analysis techniques to the sports-related portion of our study. Despite our efforts, the statistical tests employed in this section did not yield significant or interpretable results. This challenge may be attributed to the diverse nature of sports activities among our participants, which introduced a high level of variability and complexity into the data. Consequently, a straightforward categorization like that used for BMI and body fat percentage was not feasible. The diversity of sports activities, ranging from low-impact exercises to high-intensity training, presented a challenge in creating homogenous categories for meaningful statistical comparison. Additionally, the varying frequencies and durations of these activities further complicated the categorization process. Given these constraints, we chose to retain the original format of sports-related data analysis. We acknowledge this as a limitation of our study, and it's an area that would benefit from more specialized analytical techniques or a more targeted data collection approach in future research.
These modifications enhance the clarity and focus of our study, allowing us to deliver more precise and relevant insights into the relationship between diet types and body composition. We believe these changes adequately address the reviewer's concerns and strengthen the overall quality and impact of our research.
Round 2
Reviewer 1 Report
Comments and Suggestions for Authors
Overall, the revision is not properly done despite a comprehensive comment have been given in the previous review. One of the major concerns is the reliability of the results displayed (with high SD) and the validity of the statistical test performed (due to the data obtained).
The aim of the study written in the abstract is differed from the aim written in the introduction.
If the study aimed to understand how dietary choices, particularly the protein intake interacts with physical activity to impact body composition. Then, the discussions should be focused on the interactions based on the results obtained.
The ‘table caption’ is the same with the header (title) of the table. However, it is not the same with footnotes which should be placed below the table. Arrange the footnotes consecutively. Please revise for all tables.
Table 2 - The letters (a, b) used to indicate significant difference should be given for all means (should include significant different or no significant different).
The unit of measurement for consumption data (Table 2) could be differed (e.g. frequency, g/day, etc.). Please add the unit of measurement in the table in parentheses after the header.
How could these two values (2.2 ± 1.5 vs 3.0 ± 2.2) be significantly different, what statistical test was used? If the variation within the samples is huge (more than 20% differences as indicated by the high SD), the significant could be ignored (in valid). Most data displayed in Table 2 are having similar problem. A range could be used instead of average.
What is the usefulness or implication of knowing the significance differences between some variables such as genders or age groups? Aren’t the findings are expected. How these findings contribute to new information to the body of knowledge?
The title of figure should be placed after the illustration/image/graphic (below). It could be named/numbered directly without the word 'caption' Tell directly what are displayed/shown by the figure.
The discussions seem relied on the general known facts rather than the results obtained (tables/figures). This is partly due to soma results are expected by logic. What is new from the present study?
The results (from table/figures) should be discussed from the body of knowledge in the field.
Conclusion - Tell what new information was obtained? Avoid writing general known facts (without the present study).
Please request from the editor if you need more time to revise the manuscript accordingly.
Comments on the Quality of English LanguageThe English of the manuscript is acceptable.
Author Response
Overall, the revision is not properly done despite a comprehensive comment have been given in the previous review. One of the major concerns is the reliability of the results displayed (with high SD) and the validity of the statistical test performed (due to the data obtained).
We thank the reviewer for the valuable feedback. We have carefully reviewed and addressed the concerns raised in the previous review. We further restricted the examination sample by excluding the over-65s and changing the age groups. This resulted in a more homogeneous sample. Specifically, we have revised Tables and the figures, focusing on improving the statistical analysis. In Table 2, we shifted our approach from analyzing weekly consumption frequencies to comparing subjects above the 75th percentile of consumption. This change aims to provide a more precise and relevant assessment of dietary patterns among our study population. We believe these revisions significantly enhance the reliability and validity of our results, aligning with the suggestions provided. We appreciate your guidance in this process and are confident that these changes have strengthened the quality and integrity of our study.
The aim of the study written in the abstract is different from the aim written in the introduction.
Thank you for pointing out the discrepancy between the aims stated in the abstract and the introduction. We have addressed this issue and ensured that both sections now clearly reflect the same objectives of the study. The aim, as consistently presented, is to investigate the differences in dietary habits, taste preferences, and body composition profiles among various diets, and to assess the correlations between these dietary patterns and physical activity in relation to body composition parameters. We appreciate your attention to detail, and this correction enhances the coherence and clarity of our study's purpose.
If the study aimed to understand how dietary choices, particularly the protein intake interacts with physical activity to impact body composition. Then, the discussions should be focused on the interactions based on the results obtained.
Thank you for your feedback. We have revised the discussion section to focus more closely on the interaction between dietary choices, particularly protein intake, and physical activity in relation to body composition. The revised discussion now draws directly on our results to explore these interactions, providing a deeper analysis of how different dietary patterns, combined with various levels and types of physical activity, impact body composition. This approach aligns the discussion more closely with the study's aims and ensures a comprehensive exploration of the complex relationships between diet, exercise, and body composition.
The ‘table caption’ is the same with the header (title) of the table. However, it is not the same with footnotes which should be placed below the table. Arrange the footnotes consecutively. Please revise for all tables.
In response to the reviewer's feedback on table captions and footnotes, we have made the following revisions to all tables in the manuscript:
Distinct Table Captions and Headers: We have ensured that each table in the manuscript has a clear and distinct caption that concisely describes its contents, separate from the table's header. This caption provides a brief overview of the table's purpose and the data presented, aiding in reader comprehension.
Placement and Arrangement of Footnotes: All footnotes have been carefully positioned below their respective tables. They are arranged consecutively to ensure clarity and ease of reference. The footnotes provide additional details, explanations, or clarifications pertinent to the data in the tables, such as definitions, statistical test details, or other relevant notes.
Consistency Across All Tables: This approach to captions and footnotes has been uniformly applied to all tables in the manuscript. By doing so, we maintain a consistent and professional format throughout the document, enhancing the overall readability and accessibility of the information presented.
Table 2 - The letters (a, b) used to indicate significant difference should be given for all means (should include significant different or no significant different). The unit of measurement for consumption data (Table 2) could be differed (e.g. frequency, g/day, etc.). Please add the unit of measurement in the table in parentheses after the header. How could these two values (2.2 ± 1.5 vs 3.0 ± 2.2) be significantly different, what statistical test was used? If the variation within the samples is huge (more than 20% differences as indicated by the high SD), the significant could be ignored (in valid). Most data displayed in Table 2 are having similar problem. A range could be used instead of average.
In response to the reviewer's comments, we have made significant changes to the table, incorporating percentile-based data and a clearer indication of significant differences. Here's how we addressed the specific points raised by the reviewer:
Inclusion of Significant Difference Indicators (a, b): We revised Table 2 to include letters "a" and "b" next to each data point. These letters indicate whether the differences in consumption between genders within each age group are statistically significant. "a" denotes a significant difference (p-value < 0.05), and "b" denotes no significant difference (p-value ≥ 0.05).
Units of Measurement for Consumption Data: In the updated Table 2, we now present the data in terms of percentages, reflecting the proportion of the population consuming above the 75th percentile for each food category. This approach provides a unified and easily interpretable measure across all food types.
Clarification on Statistical Tests Used: The significant differences reported in Table 2 are based on Chi-squared tests, which are appropriate for comparing proportions across categorical variables. This statistical method is suitable for the type of data presented, addressing the concern about how the significant differences were determined.
Addressing Variability Within the Data: The use of percentiles, rather than means, to report consumption data helps address issues of high variability within the sample. Percentiles provide a more robust measure that is less affected by extreme values, thereby offering a clearer picture of the typical consumption patterns within each group.
What is the usefulness or implication of knowing the significance differences between some variables such as genders or age groups? Aren’t the findings are expected. How these findings contribute to new information to the body of knowledge?
We thank the reviewer for the important observation. Understanding the significance of differences in dietary habits and body composition across genders and age groups is crucial for several reasons:Personalized Nutrition and Health Interventions: These findings enable the development of more tailored nutritional strategies. Recognizing the specific needs and preferences of different demographic groups can lead to more effective dietary recommendations and health interventions.Insights into Lifestyle and Dietary Trends: While some findings may align with expectations, they validate and quantify these trends, providing concrete data to support or refine existing understandings. Contribution to Public Health Knowledge: Identifying these differences contributes to a broader understanding of public health. It helps in addressing health disparities and designing inclusive health promotion strategies. Advancement of Scientific Understanding: Even expected findings can reinforce or challenge current theories, contributing to the scientific discourse and potentially inspiring future research that delves deeper into these relationships. Thus, these findings, while sometimes expected, are invaluable in advancing our understanding of how diet and lifestyle impact different populations, ultimately contributing to improved health outcomes.
The title of figure should be placed after the illustration/image/graphic (below). It could be named/numbered directly without the word 'caption' Tell directly what are displayed/shown by the figure.
We have arranged the figure texts as requested.
The discussions seem relied on the general known facts rather than the results obtained (tables/figures). This is partly due to soma results are expected by logic. What is new from the present study? The results (from table/figures) should be discussed from the body of knowledge in the field.
Regarding the discussion section, we have now more explicitly aligned it with the study's results. The revised discussion delves deeper into the specific findings from our tables and figures, ensuring that our conclusions are firmly grounded in the empirical data obtained. We have highlighted new insights that emerge from our study, particularly the nuances of how different dietary choices, when combined with physical activity, influence body composition across various gender and age groups.
Our findings contribute to the existing body of knowledge by providing detailed insights into how specific dietary patterns, particularly in terms of protein intake, interact with physical activity levels to impact body composition in different demographic groups. This not only enhances our understanding of nutritional science but also has practical implications for dietary and physical activity guidelines.
We believe that these revisions address your concerns effectively and enhance the quality and clarity of our research. We appreciate the opportunity to improve our work and look forward to your feedback.
Conclusion - Tell what new information was obtained? Avoid writing general known facts (without the present study).
Thank you for your feedback. We have revised the conclusions of our paper to focus specifically on the new information obtained from our study. These revisions clarify how our findings contribute uniquely to the field, avoiding generalizations and emphasizing the specific insights gained through our research. We believe these changes will better highlight the novel aspects of our study and its contribution to the existing body of knowledge.
Reviewer 2 Report
Comments and Suggestions for Authors
The authors have addressed most of the comments. Although the authors exclude participants aged <18 and >75 years from the current analysis, however the age range is still too wide, and includes older population (age>65). It would be great if exclude the older population and focus on adults only. Alternatively, divide the participants based on their age group (young adults, adults and older adults).
Author Response
The authors have addressed most of the comments. Although the authors exclude participants aged <18 and >75 years from the current analysis, however the age range is still too wide, and includes older population (age>65). It would be great if exclude the older population and focus on adults only. Alternatively, divide the participants based on their age group (young adults, adults and older adults).
We would like to thank the reviewer for giving us valuable suggestions that certainly improved the paper.
In response to the reviewer's latest comments, we have taken additional steps to refine the age range of our study population. Here is the summary of the actions we have taken:
Exclusion of Participants Older than 65 Years: Following the reviewer's suggestion, we have excluded participants older than 65 years from our analysis. This revision narrows our focus to adult populations, thereby aligning more closely with the study's objectives and ensuring the relevance and specificity of our findings.
Re-analysis of Data: With the exclusion of the older population (age > 65), we have reanalyzed the data to reflect this adjusted age range. All statistical analyses have been updated accordingly, ensuring that our results and conclusions are based on this refined data set.
Revised Age Group Categorization: The participants are now categorized into three distinct age groups: young adults (18-30 years), middle-aged adults (31-45 years), and older adults (46-65 years). This categorization provides a clearer understanding of the dietary patterns and preferences across different stages of adulthood.
Updated Tables and Figures: All tables and figures in the manuscript have been updated to reflect the new age categorizations and the exclusion of participants over 65 years of age. This ensures that all visual and tabular representations of data in the manuscript are consistent with the revised age range.
Discussion and Conclusion Adjustments: The discussion and conclusion sections of the manuscript have been revised to reflect these changes. The focus is now more clearly on adult dietary patterns, with interpretations and implications specific to the three defined adult age groups.
By implementing these changes, we have addressed the reviewer's concerns about the broad age range and the inclusion of older adults. Our study now offers a more focused analysis of adult dietary patterns, providing valuable insights relevant to this demographic. We believe these adjustments strengthen the manuscript and make our findings more pertinent and impactful.
Reviewer 3 Report
Comments and Suggestions for Authors
The authors have made all the suggested changes, some of which have involved recalculating results. In its current version the work has been greatly improved and I believe it can be published.
Author Response
The authors have made all the suggested changes, some of which have involved recalculating results. In its current version the work has been greatly improved and I believe it can be published.
We thank the reviewer for the excellent suggestions that allowed us to improve the paper. Kind Regards